# Relationship between Middle Cerebral Artery Pulsatility Index and Delayed Neurocognitive Recovery in Patients undergoing Robot-Assisted Laparoscopic Prostatectomy

**DOI:** 10.3390/jcm12031070

**Published:** 2023-01-30

**Authors:** Paola Aceto, Andrea Russo, Claudia Galletta, Chiara Schipa, Bruno Romanò, Ersilia Luca, Emilio Sacco, Angelo Totaro, Carlo Lai, Marianna Mazza, Bruno Federico, Liliana Sollazzi

**Affiliations:** 1Dipartimento di Scienze dell’Emergenza, Anestesiologiche e della Rianimazione, Fondazione Policlinico Universitario A. Gemelli IRCCS, 00168 Rome, Italy; 2Dipartimento di Scienze Biotecnologiche di Base, Cliniche Intensivologiche e Perioperatorie, Università Cattolica del Sacro Cuore, 00168 Rome, Italy; 3Department of Urology, Fondazione Policlinico Universitario A. Gemelli IRCCS, 00168 Rome, Italy; 4Institute of Urology, Universita Cattolica del S. Cuore-Fondazione Policlinico A. Gemelli, 00168 Rome, Italy; 5Department of Dynamic and Clinical Psychology and Health Studies, Sapienza University, 00185 Rome, Italy; 6Institute of Psychiatry and Psychology, Department of Geriatrics, Neuroscience and Orthopedics, Fondazione Policlinico Universitario A. Gemelli IRCCS, 00168 Rome, Italy; 7Department of Psychiatry, Università Cattolica del Sacro Cuore, 00168 Rome, Italy; 8Department of Human Sciences, Society and Health, University of Cassino and Southern Lazio, 03043 Cassino, Italy

**Keywords:** transcranial doppler, postoperative cognitive dysfunction, robotic-assisted prostatectomy

## Abstract

A steep Trendelenburg (ST) position combined with pneumoperitoneum may cause alterations in cerebral blood flow with the possible occurrence of postoperative cognitive disorders. No studies have yet investigated if these alterations may be associated with the occurrence of postoperative cognitive disorders. The aim of the study was to evaluate the association between an increased middle cerebral artery pulsatility index (Pi), measured by transcranial doppler (TCD) 1 h after ST combined with pneumoperitoneum, and delayed neurocognitive recovery (dNCR) in 60 elderly patients undergoing robotic-assisted laparoscopic prostatectomy (RALP). Inclusion criteria were: ≥65 years; ASA class II–III; Mini-Mental Examination score > 23. Exclusion criteria were: neurological or psychiatric pathologies; any conditions that could interfere with test performance; severe hypertension or vascular diseases; alcohol or substance abuse; chronic pain; and an inability to understand Italian. dNCR was evaluated via neuropsychological test battery before and after surgery. Anesthesia protocol and monitoring were standardized. The middle cerebral artery Pi was measured by TCD, through the trans-temporal window and using a 2.5 MHz ultrasound probe at specific time points before and during surgery. In total, 20 patients experiencing dNCR showed a significantly higher Pi after 1 h from ST compared with patients without dNCR (1.10 (1.0–1.19 95% CI) vs. 0.87 (0.80–0.93 95% CI); *p* = 0.003). These results support a great vulnerability of the cerebral circulation to combined ST and pneumoperitoneum in patients who developed dNCR. TCD could be used as an intraoperative tool to prevent the occurrence of dNCR in patients undergoing RALP.

## 1. Introduction

It is known that a steep Trendelenburg (ST) position, especially when combined with pneumoperitoneum, can cause alterations of brain regulatory mechanisms [1] that, in elderly patients, may lead to the onset of postoperative cognitive dysfunction (pCD) [2]. At present, there are no studies investigating the possible relationship between changes in cerebral blood flow, caused by ST combined with pneumoperitoneum, and the occurrence of postoperative cognitive disorders.

Transcranial Doppler (TCD) ultrasonography allows repeated, non-invasive investigations of rapid changes in intracerebral perfusion by assessing middle cerebral artery flow. The most commonly used hemodynamic index is the Gosling pulsatility index (Pi) [3] which has traditionally been interpreted as a descriptor of non-invasive intracranial pressure (ICP) in brain injury as well as in the normal brain [4,5].

The main objective of this study was to evaluate the association between a higher Pi at 1 h from the onset of ST combined with pneumoperitoneum and the occurrence of delayed neurocognitive recovery (dNCR). The association between dNCR and emergence agitation (EA) or postoperative delirium (POD) was also explored.

## 2. Materials and Methods

This single-center, prospective study was approved by the local Institutional Ethic Committee (ID 1781). Written informed consent was obtained from each patient before the study. All patients scheduled for robotic-assisted laparoscopic prostatectomy (RALP) were screened for enrolment. Patients aged ≥65 years with an ASA physical status classification class II–III and a Mini-Mental Examination (MMSE) score (corrected for age and educational level) of >23 were included. Patients who refused to participate, with known neurological or psychiatric diseases, under chronic psychiatric drugs or other conditions that could interfere with test performance (e.g., blindness and deafness), a history of severe hypertension, a significant carotid or cerebral vascular disease, alcohol or substance abuse, chronic pain, and an inability to understand the Italian language were excluded.

### 2.1. Anesthesia Protocol

All patients underwent standard monitoring: electrocardiogram, non-invasive arterial blood pressure, pulse oximetry, expiratory gas concentration, bispectral index (BIS), and diuresis. Anesthesia was induced with fentanyl 3 µg/kg, propofol 2 mg/kg, whilst tracheal intubation was facilitated by the administration of rocuronium 0.6 mg/kg. Anesthesia was then maintained with Sevoflurane adjusted according to the BIS value which was kept between 40 and 60. All patients were mechanically ventilated with a tidal volume of 7 mL/kg and the respiratory rate was adjusted to maintain the carbon dioxide end-tidal between 35 and 45 mmHg. Rocuronium 0.15 mg/kg was then repeated in order to keep a deep neuromuscular block (Post Tetanic Count ≤ 2). For intraoperative analgesia, remifentanil was administered in continuous infusion at concentrations varying from 0.05 to 0.25 mcg/kg/min, depending on heart rate and mean arterial pressure variations. Balanced solutions were administered at 1–5 mL/kg/h intraoperatively and 1000 mL for 24 h postoperatively.

After prostate removal, remifentanil infusion was stopped and a 2 mL/h elastomeric pump with Tramadol 400 mg in 48 mL of 0.9% NaCl solution was started [6]. For all patients, before extubation, Paracetamol 1 gr and Ketorolac 30 mg were administered.

Boluses of morphine (0.03 mg/kg; maximum dose 10 mg) were used to treat post-operative pain in the recovery room (RR), while intravenous Tramadol 100 mg was the rescue dose therapy for pain control during ward stay and was administered if the Numeric Rating Scale (NRS) value was ≥5. All patients received Paracetamol 1 gr every 8 h for the first 24 h after surgery.

### 2.2. Data Collection and Measurements

(1)For the diagnosis of dNCR, the following tests were performed on the day before surgery and on the 2nd day postoperatively: the Rey Auditory Verbal Learning Test (RAVLT), the Raven’s Progressive Matrices test, the trail-making test (part A and part B), the Clock drawing test, a phonemic and semantic verbal fluency test, and the Rey–Osterrieth complex figure test (ROCF). dNCR (dichotomous variable) was diagnosed in the individual patient when there was a postoperative decrement of ≥1 standard deviation (SD) (of the whole group at baseline) in a single test and no improvement (score ≥ 1 SD) in the other tests [7]. An improvement in a test score—between the first and the second assessment—smaller than 1 SD of the whole group at baseline was interpreted as a consequence of the practice effect [8].(2)The onset of POD was assessed by the Confusion Assessment Method (CAM). CAM was administered in the RR and daily until discharge [9].(3)Anxiety and depression were also assessed on the day before surgery using the State-Trait Anxiety Inventory (STAI) [10] and the Beck Depression Inventory Second Edition (BDI-II) [11], respectively.(4)During surgery, mean blood pressure, heart rate, BIS, carbon dioxide end-tidal, and pneumoperitoneum-pressure values were recorded (when applicable): before (T1) and after (T2) the induction of anesthesia; thirty minutes (T3) and one hour after the start of ST combined with pneumoperitoneum (T4); before ST removal (T5); ten minutes after the end of ST and pneumoperitoneum before waking up (T6). ST, a position used routinely during RALP, involves lowering (by 45 degrees) the top of the operating table from the head side and maintaining this position for almost the entire duration of the surgery. Pneumoperitoneum pressure was applied immediately before ST application and maintained at values < 12 mmHg.

With appropriate equipment (Hitachi) and a 2.5 MHz ultrasound probe, the Trans Cranial Doppler (TCD) was performed through the trans-temporal window—located in the middle point between the tragus and the external angle of the ipsilateral eye—at all the time points listed above. The middle cerebral artery (MCA) which is located approximately 30–60 mm deep, and its flow, approaching the probe, appears as a positive wave. Pi was measured according to Gosling’s method [12]. Resistivity index (Ri) was also assessed [11].

(5)Pain was assessed using NRS ranging from 0 with no pain to 10 with the worst pain ever felt at the following times: at the patient’s arrival in the recovery room, and after 1, 2, 8, 12, 24 and 48 h.(6)The 36-Item Short Form Health Survey (SF-36) was assessed on the day before surgery and on the 2nd day postoperatively [13].

The following data were also recorded: demographic parameters (age, Body Mass Index, years of education); risk stratification variables (ASA physical, status, Charlson Comorbidity Index); duration of surgery and anesthesia; amount of infused balanced solution; ST duration; diuresis; remifentanil consumption; morphine use in the recovery room and tramadol administration in the ward and hospital stay.

### 2.3. Statistical Analysis

The a priori power analysis for the calculation of the sample size was based on differences between means (power analysis with Student’s unpaired *t*-test) and performed with G*Power 3.1.5 software. Considering a 25% incidence of dNCR and a 50% difference in Pi after pneumoperitoneum + ST between patients with and without dNCR (effect size d = 1), a minimum of 52 patients was considered necessary by calculating an allocation ratio of 4/1 for a two-tailed test with β = 0.80 and α = 0.05. A total of 60 patients were foreseen for enrolment to deal with any dropouts.

Clinical and demographic characteristics were indicated using descriptive statistics. Quantitative variables were described using the mean and a 95% confidence interval (CI). The qualitative variables were summarized using absolute values. The *t*-test for continuous variables and Yates corrected chi-square for dichotomous or discrete variables were used to evaluate the differences between patients with and without dNCR. Furthermore, repeated measures ANOVA with Bonferroni correction was performed for variables assessed at different times. Logistic regression was used to identify possible dNCR predictors, including only variables significant at univariate analysis. The cut-off of significant predictors was calculated using non-parametric ROC (Receiver Operating Characteristic) analysis and establishing a sensitivity ≥ 0.8. The data were analyzed using the Statistica software (version 8.0) or STATA (version 14.0).

## 3. Results

Sixty-three patients were assessed for eligibility, three of whom were excluded for the reasons shown in Figure 1 and sixty were finally enrolled. Patients’ characteristics and intraoperative data are shown in Table 1. No differences were found for demographic and anesthesia/surgery variables (Table 1).

Of the 60 enrolled patients, 11 experienced emergence agitation upon awakening in the operating room, 3 had POD in the recovery room, and 20 patients were diagnosed with dNCR by the assessment on the 2nd postoperative day (*n* = 17).

All three patients who experienced POD were subsequently diagnosed with dNCR, showing a statistically significant association between POD and dNCR (*p* = 0.03).

The ANOVA results showed a significant effect of the Group per Time interaction (F (5290) = 2.35; *p* = 0.04) for Pi. In the group of patients with dNCR, a significantly higher Pi at t4 was found compared with patients without dNCR (1.10 (1.0–1.19 95% CI) vs. 0.87 (0.80–0.93 95% CI); *p* = 0.003) (Figure 2). The increase in Pi at 1 h after ST compared to the values after anesthesia induction was significantly higher in the dNCR group (0.12 ± 0.25 vs. −0.05 ± 0.15; t: −3.34; *p* = 0.008). The other variables measured during (mean arterial pressure, heart rate, end-tidal CO_2_, BIS, pneumoperitoneum) and after anesthesia including NRS (see Appendix A), as well as morphine and tramadol consumption (see Table 1), were comparable between the two groups. While the ANOVA showed a significant effect of the interaction Group per Time (F (1290) =3.00; *p* = 0.01) for Ri (see Appendix A), no significant differences were found between the two groups at post hoc analyses. Logistic regression showed that Pi at 1 h after ST (*p* = 0.002) was a predictor of dNCR (Likelihood Ratio chi^2^ = 13.26; *p* = 0.003). Moreover, a Pi of 0.9 was identified as the determinant cut-off for dNCR (sensitivity = 80.0%; specificity = 65.0%; AUC: 0.76).

The significantly altered neuropsychological tests after surgery were the Rey Auditory Verbal Learning Test (RAVLT) and the Rey–Osterrieth complex figure (ROCF) test recall (see Table 2).

As regards health status, there was a significant reduction in both “Emotional Well-being” (*p* < 0.0001) and “Energy-Fatigue” (*p* = 0.0008) items on the 2nd day postoperatively compared to basal values (before surgery) only in patients with dNCR. Moreover, the “Emotional well-being” item was significantly lower in patients with dNCR compared to those without dNCR (*p* = 0.028) (see Appendix A).

## 4. Discussion

Our results show that one-third of the population studied was diagnosed with dNCR. These data are confirmed in the literature which shows a higher incidence in elderly patients [14]. In this study, we found a statistically significant association between a higher value of Pi one hour from the start of ST combined with pneumoperitoneum and the onset of dNCR. Pi measured at this time point was chosen as the main variable as we hypothesized that time length in ST contributes to altering cerebral hemodynamics.

The combination of pneumoperitoneum and ST can increase ICP as proven by ultrasonographic measurement of Pi [1] or optic nerve sheath diameter [15]. In our study, the increase in Pi 1 h after ST compared to that after anesthesia induction (before the application of ST and Trendelenburg) was significant in patients with dNCR, reinforcing this interpretation. Kalmar et al., also hypothesized that venous congestion due to Trendelenburg was the main determinant of the increase in ICP [16].

Based on our results, Trendelenburg degrees could be reduced until the Pi drops below the cut-off of 0.9. Maintaining a stable ultralow pneumoperitoneum pressure using a valveless insufflation system could be another possible protective strategy to avoid the occurrence of dNCR [17].

The association between cerebral hemodynamics and cognitive outcome after anesthesia has been poorly investigated. Chen et al., indirectly measured the ICP during RALP through the variation of the ONSD and found a potential indirect link between the increase in ICP during the combination of pneumoperitoneum and ST and the onset of short-term cognitive disorders [2]. However, Chen et al., reported, among the limitations of their study, the use of MMSE [2], which is a screening test for of evaluating cognitive impairment in older adults while dNCR needs a battery of more specific tests to be diagnosed [7]. On the contrary, Goettel et al. and Kim et al. [14] showed that impaired intraoperative cerebral autoregulation seems not to be predictive of dNCR in elderly patients after major non-cardiac surgery [18].

The relationship between the increase in ICP determined by ST and pneumoperitoneum with dNCR has never been systematically investigated. However, one of the pathophysiological hypotheses of pCD concerns cerebral hypoperfusion during surgery. This variation could be the epiphenomenon of a greater vulnerability of these patients to the venous congestion caused by ST with consequent hypoperfusion and reduced metabolic oxygen supply. However, brain oxygenation was not measured in our study, thus this cause–effect relationship cannot be confirmed.

Moreover, a previous review on the onset of pCD in non-cardiac surgery highlighted that the correlation between pCD and intraoperative cerebral hypoxemia is not so strong and more limited to inflammatory mediators triggered by stress caused by anesthesia, surgery, and hospitalization [19]. The inflammation mediators could also have had an essential role in dNCR pathophysiology in our study due to a possibly elevated permeability of the blood–brain barrier in patients with dNCR [20]. Thus, the concomitant increase in ICP with consequent stasis at the cerebral level in susceptible patients could have facilitated the entrance of cytokines in the brain during surgical stress causing neuroinflammation.

Among the other risk factors for the onset of postoperative cognitive disorders, monitoring of the depth of anesthesia was discussed in several papers as a preventive measure [21,22]. Chan et al., have shown that a guided BIS anesthesia reduced the occurrence of both short and long-term pCD and POD [23]. Similarly, Kotekar et al., argued that intraoperative monitoring of the depth of anesthesia, especially in older patients, can help reduce the onset of pCD [24]. However, in our study, the mean BIS did not differ between patients with and without dNCR.

An important limitation of the study is the lack of long-term follow-up to assess either the persistence of cognitive dysfunction or the impact of pCD on quality of life over time. It would be interesting for future studies to investigate if these changes may also occur in long-lasting surgeries such as robotic cystectomy.

Another limitation is that Pi was considered an indirect index of the ICP, even if this issue deserves further investigation. However, the absence of significant differences in Ri between patients with and without dNCR in our study could confirm that, when compared with the PI, the RI index is less sensitive to ICP variations [25]. Moreover—even though expired CO_2_ did not show significant changes—we cannot exclude an influence of the PaCO^2^ increase during pneumoperitoneum [26] on Pi, which may increase in response to hypercapnia.

In conclusion, the most relevant result of this study is the association between the increase in Pi after one hour from ST under pneumoperitoneum and dNCR. These results support a great vulnerability of the cerebral circulation to the ST combined with pneumoperitoneum in patients who develop dNCR. Even if further studies are needed to confirm these findings, middle cerebral artery Pi could be used as a prognostic indicator of an unfavorable cognitive outcome and constitutes a deterrent to modifying the perioperative therapeutic strategy in patients with risk factors for dNCR. The findings of the present study had relevant clinical implications for the chance to predict and prevent dNCR and the consequent impairment in quality of life after surgery.

## Figures and Tables

**Figure 1 jcm-12-01070-f001:**
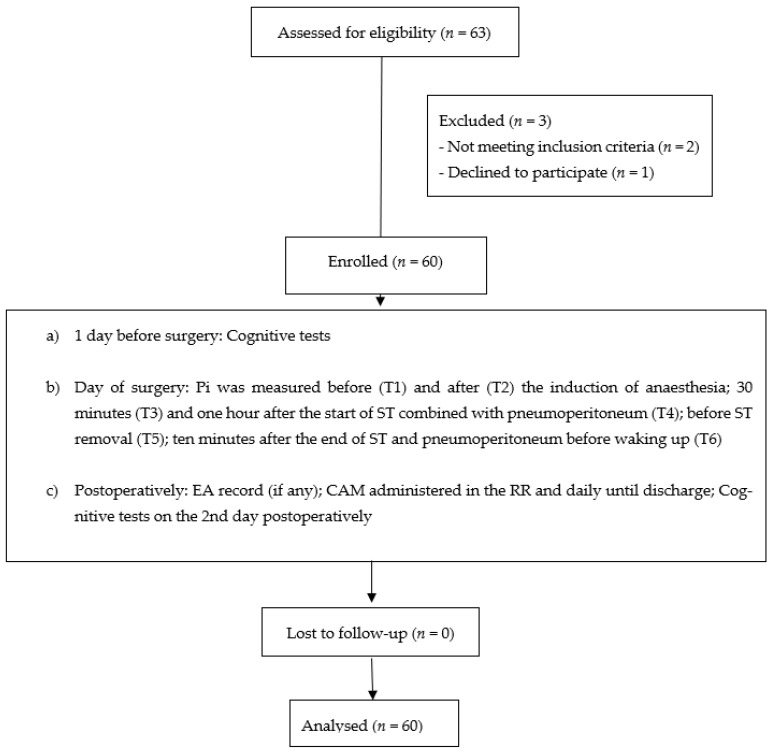
Study diagram flow with the description of study design. ST, steep Trendelenburg; Pi, pulsatility index; EA, emergence agitation; CAM, Confusion Assessment Method; RR, recovery room.

**Figure 2 jcm-12-01070-f002:**
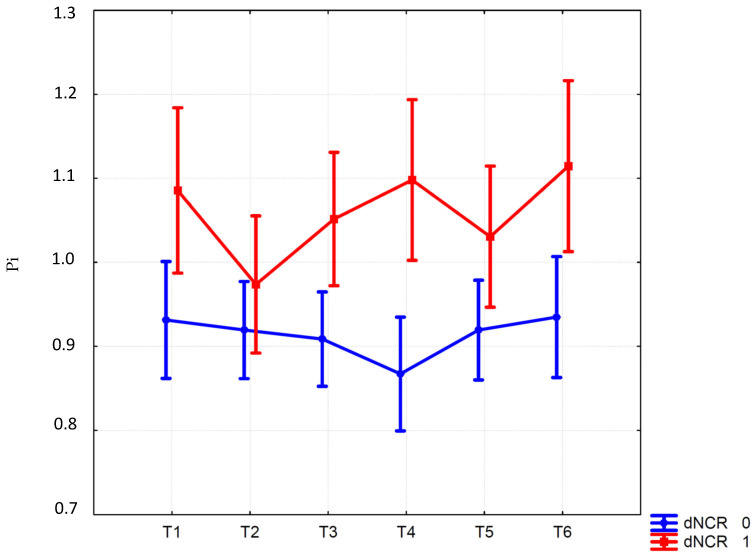
Pulsatility index (Pi) at different time points: baseline (T1), after the induction of anesthesia (T2); 30 min after the start of the Trendelenburg position (T3); one hour from the start of ST (T4), before ST removal (T5); before waking up, ten minutes after the end of ST (T6). Vertical bars denote 95% CI.

**Table 1 jcm-12-01070-t001:** Main pre-, intra-, and postoperative parameters in patients with and without dNCR. Values are means (95% confidence intervals) or numbers. BMI: Body Mass Index; ASA: American Society of Anesthesiologists physical status classification; CCI: Charlson Comorbidity Index; MMSE: mini-mental state examination score corrected for age and educational level; STAI: State-Trait Anxiety Inventory; Y1: state anxiety; Y2: trait anxiety; BDI-II: Beck Depression Inventory Second Edition; ST: Steep Trendelenburg; I.O.: intraoperative; P.O.: postoperative; EA: emergence agitation; POD: postoperative delirium.

	Patients without dNCR (*n* = 40)	Patients with dNCR (*n* = 20)	t or χ2 (df)	*p*
Age, years	69.6 (68.6–70.6)	70.6 (68.4–72.6)	−1.04 (58)	0.30
BMI, kg/m^2^	26.2 (25.3–27.2)	25.7 (24.4–27.0)	0.72 (58)	0.47
ASA, II/III	38/2	19/1	0.39 (2)	0.53
CCI	4.6 (4.3–4.8)	4.7 (4.3–5.0)	−0.63 (58)	0.53
MMSE	25.3 (24.2–26.5)	26.0 (25.5–26.4)	1.28 (58)	0.21
STAI-Y1	34.7 (30.8–38.6)	35.5 (30.2–40.8)	−0.24 (58)	0.80
STAI-Y2	31.3 (29.4–33.2)	33.1 (29.0–37.3)	−0.93 (58)	0.35
DBI-II	7.6 (5.9–9.4)	8.7 (5.4–12.0)	−0.67 (58)	0.50
Balanced solution, ml	620.0 (535.8–704.2)	605.0 (448.1–761.9)	0.19 (58)	0.85
Surgery duration, min	178.1 (162.7–193.6)	176.8 (158.0–195.7)	0.10 (58)	0.92
Anesthesia duration, min	207.2 (190.8–223.6)	208.2 (188.2–228.2)	−0–07 (58)	0.94
ST duration, min	145.3 (132.1–158.5)	150.1 (134.2–165.9)	−0.44 (58)	0.66
Diuresis, ml	234.7 (200.7–268.8)	303.0 (208.7–397.3)	−1.72 (58)	0.09
I.O. Remifentanil, mcg	745.5 (621.9–869.1)	984.7 (708.6–1260.9)	−1.89 (58)	0.06
P.O. Morphine, Yes/No	5/35	4/16	0.15 1)	0.70
P.O. Tramadol, Yes/No	13/27	6/14	0.01 (1)	0.92
EA, Yes/No	5/35	6/14	1.68 (1)	0.19
POD, Yes/No	0/40	3/17	3.55 (1)	0.03
Hospital stay (days)	5.1 (4.7–5.5)	5.7 (4.4–6.9)	−1.06 (58)	0.29

**Table 2 jcm-12-01070-t002:** Neuropsychological tests (difference between postoperative and preoperative values) in patients with and without dNCR. RAVLT, Rey Auditory Verbal Learning Test; stm, short-term memory; ltm, long-term memory; re, recency effect; PPMT, Raven’s Progressive Matrices test; TMT, trail making test; CDT, clock drawing test; pVFT, phonemic verbal fluency test; sVFT, semantic verbal fluency test; ROCF, Rey–Osterrieth complex figure test. Values are means (95% confidence intervals).

	Patients without dNCR (*n* = 40)	Patients with dNCR (*n* = 20)	t (df = 58)	*p*
RAVLT, stm	1.3 (0.6–2.1)	0.1 (−0.8–1.0)	2.03	0.04
RAVLT, ltm	4.2 (3.2–5.1)	1.4 (−0.2–3.1)	3.19	0.002
RAVLT, re	1.5 (0.9–2.2)	−0.4 (−1.3–0.5)	3.55	0.0007
RPMT	−0.01 (−0.8–0.8)	0.09 (−1.3–1.5)	−0.14	0.89
TMT-A	8.7 (−1.8–19.2)	−4.8 (−20.1–10.3)	1.51	0.14
TMT-B	3.7 (−13.6–21.0)	−2.6 (−30.3–25.1)	0.41	0.68
CDT	−0.3 (−0.6–0.01)	0.05 (−0.6–0.7)	−1.12	0.27
pVFT	0.6 (−1.2–2.5)	−0.14 (−3.4–3.1)	0.46	0.65
sVFT	0.8 (−0.5–2.2)	0.16 (−1.5–1.9)	0.63	0.53
ROCF, copy	−2.8 (−5.2–−0.4)	−3.5 (−5.5–−1.6)	0.41	0.68
ROCF, recall	4.6 (3.0–6.3)	1.2 (−0.9–3.2)	2.60	0.01

## Data Availability

Data presented in this study are available upon request.

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
