# Peer review of "Relationship between Middle Cerebral Artery Pulsatility Index and Delayed Neurocognitive Recovery in Patients undergoing Robot-Assisted Laparoscopic Prostatectomy"

_jcm, 2023, doi:10.3390/jcm12031070_

Round 1

Reviewer 1 Report

In this manuscript, the authors focused on the relationship between cerebral hemodynamic changes and the alteration of perioperative neurocognitive function in the patients undergoing robot-assisted laparoscopic prostatectomy (RALP) with Steep Trendelenburg combined with pneumoperitoneum. According to the result, the authors observed that patients experiencing dNCR showed a significantly higher Pi after 1h from ST compared with patients without dNCR. The authors attempted to provide an indicator for the prediction of postoperative neurocognitive dysfunction. Here, there are still some contents that need to be explained with more detail.

1.      In “Data collection and measurements”, why did you determine “postoperative decrement of ≥1 standard deviation in a single test and no improvement in the other tests” as the diagnostic criteria? As we known, different paradigms of cognitive tests reflected different cognitive categories. If some scores increase, will it cover up dNCR? Please confirm.

2.      I suggest to add some quantitative correlative analysis to present the correlation between dNCR and the increase of Pi at 1 hour after ST

3.      In Discussion, I suggest to further analyze the possibility of delayed neurocognitive recovery caused by increased cerebral blood flow mediated by combined ST and pneumoperitoneum. For instance, the vascular permeability, BBB, oxygen supply and etc.

Reviewer 2 Report

The authors present a very interesting study about the impact of longer Trendelenburg position. I have the following comments:

1. It would be interesting to know, how long the dNCR persists in these patients.

2. Does the OR-time has any impact on dNCR?

3. What should be the cut-off of PI in these patients.

4. What should be the consequences? Convert the operation to open or reducing Trendelenburg until the PI drops?

5. What could be the influence of lower abdominal pressure by use of the AirSeal device? Could this compensate some how the Trendelenburg.

6. Finally, in my clinical practice, I am not aware of significant cognitive changes in my patients after RALP. This was different in long-lasting surgeries like robotic cystectomy. It would be interesting to monitor this patient group rather than patients undergoing RALP.

Round 2

Reviewer 2 Report

NOne